# Epidemiology, Diagnosis, Staging and Multimodal Therapy of Esophageal and Gastric Tumors

**DOI:** 10.3390/cancers13030582

**Published:** 2021-02-02

**Authors:** Donelle Cummings, Joyce Wong, Russell Palm, Sarah Hoffe, Khaldoun Almhanna, Shivakumar Vignesh

**Affiliations:** 1Division of Gastroenterology and Hepatology, Department of Medicine, New York Medical College, New York City Health and Hospitals Corporation-Metropolitan Hospital Center, 1901 First Avenue, New York, NY 10029, USA; cummingd2@nychhc.org; 2Division of Surgery, Mid Atlantic Kaiser Permanente, 700 2nd St. NE, 6th Floor, Washington, DC 20002, USA; Joyce.x.wong@kp.org; 3Department of Radiation Oncology, Moffitt Cancer Center, 12902 USF Magnolia Drive, Tampa, FL 33612, USA; russell.palm@moffitt.org (R.P.); sarah.hoffe@moffitt.org (S.H.); 4Division of Hematology/Oncology, Lifespan Cancer Institute, Rhode Island Hospital, The Warren Alpert Medical School of Brown University, 593 Eddy St, George 312, Providence, RI 02903, USA; Kalmhanna@lifespan.org; 5Division of Gastroenterology and Hepatology, Department of Medicine, SUNY Downstate Health Sciences University, MSC 1196, 450 Clarkson Avenue, Brooklyn, NY 11203, USA

**Keywords:** esophageal cancer, gastric cancer, gastrointestinal stromal tumor, neuroendocrine tumor, MALT lymphoma, mucosal resection, submucosal dissection

## Abstract

**Simple Summary:**

Upper gastrointestinal tumors involve the tubular organs from the upper esophagus, the stomach, and the first part of the small intestine. Esophageal and gastric cancers are responsible for high rates of disease, morbidity, and mortality throughout the world. Diagnosis of these tumors involves a combination of clinical symptoms, endoscopy, endoscopic ultrasound, and radiological studies. Treatment depends on input from many medical doctors including gastroenterologists, surgeons, pathologists, medical oncologists, radiologists, and radiation oncologists. Treatment may include endoscopy, surgery, chemotherapy, radiation therapy, or a combination of these approaches. Future directions of diagnosis may include improvements in endoscopy, endoscopic ultrasound, blood testing, and tissue testing.

**Abstract:**

Gastric and esophageal tumors are diverse neoplasms that involve mucosal and submucosal tissue layers and include squamous cell carcinomas, adenocarcinomas, spindle cell neoplasms, neuroendocrine tumors, marginal B cell lymphomas, along with less common tumors. The worldwide burden of esophageal and gastric malignancies is significant, with esophageal and gastric cancer representing the ninth and fifth most common cancers, respectively. The approach to diagnosis and staging of these lesions is multimodal and includes a combination of gastrointestinal endoscopy, endoscopic ultrasound, and cross-sectional imaging. Likewise, therapy is multidisciplinary and combines therapeutic endoscopy, surgery, radiotherapy, and systemic chemotherapeutic tools. Future directions for diagnosis of esophageal and gastric malignancies are evolving rapidly and will involve advances in endoscopic and endosonographic techniques including tethered capsules, optical coherence tomography, along with targeted cytologic and serological analyses.

## 1. Introduction

### 1.1. Anatomic Principles

Upper gastrointestinal neoplasia is a complex disease process of the digestive organs of the foregut involving structures from the upper esophageal sphincter to the duodenum at the ligament of Treitz. These lesions originate from the mucosal or submucosal tissue layers of the esophagus, stomach, and duodenum and involve numerous cell types, including squamous cell carcinomas, adenocarcinomas, spindle-cell neoplasms, lymphomas, neuroendocrine tumors, and several less common tumors.

### 1.2. Epidemiology

Available WHO statistics indicate that upper gastrointestinal malignancies are responsible for a significant disease burden globally. Esophageal cancer (including adenocarcinoma and squamous cell carcinoma) is the ninth most common malignancy worldwide and has the sixth-highest cancer mortality [1]. Squamous cell cancer of the esophagus accounts for approximately 90% of incident esophageal cancers all over the world. It is much less common than adenocarcinoma in the United States. Gastric cancer (adenocarcinoma) represents an even greater disease burden, and is the fifth most common cancer, representing the fourth-highest cause of cancer mortality worldwide [2]. Gastric adenocarcinoma exhibits a unique geographic predilection, with a high incidence documented in Asian countries, particularly in Japan and South Korea (41 cases per 100,000 persons) and lowest in North America (below 5 cases per 100,000 persons) [3]. Gastric adenocarcinoma is subdivided into the intestinal and diffuse subgroups based on the Lauren classification, which includes intestinal, diffuse, mixed, and indeterminate histological variants. Alternatively, the WHO classification is based upon histology and subdivides gastric adenocarcinoma into tubular, papillary, mucinous, poorly cohesive, and rare variants [2,4].

Gastric lymphomas account for 1–6% of gastric neoplasms diagnosed annually. Infection with *H. pylori* is known to be an important factor in carcinogenesis. Chronic *H. pylori* infection is thought to promote clonal expansion of gastric lymphocytes, leading to gastric lymphoma. Approximately 40–50% of cases are gastric marginal zone lymphoma of mucosa-associated lymphoid tissue (MALT) and 20–40% are extra-nodal lymphomas [5].

Gastrointestinal stromal tumors (GISTs) are rare mesenchymal tumors. They may occur anywhere along the digestive tract, but they are most commonly located in the stomach and small intestine. Within the US, the incidence is estimated at 4000 to 6000 cases per year [6].

Gastrointestinal neuroendocrine tumors (NET) are uncommon and may occur anywhere along the gastrointestinal tract or within the pancreaticobiliary system. There are four subtypes of gastric neuroendocrine tumors (I, II, III, and IV). Type I tumors are associated with atrophic chronic gastritis. They may be well-differentiated and multifocal. Smaller type I lesions may be amenable to endoscopic resection and surveillance. Type II NETs are associated with Zollinger Ellison Syndrome (as part of Multiple Endocrine Neoplasia I) and have an increased malignant potential. Type III and Type IV neuroendocrine tumors have a higher malignant potential and may be metastatic on presentation. In the US, from 2000 to 2012, SEER 18 Registry data indicated a neuroendocrine tumor incidence of 3.56 per 100,000 in gastroenteropancreatic sites. It is estimated that only 7–8% of NETS are found in the stomach [6].

### 1.3. Natural History

Adenocarcinomas, squamous cell carcinomas, neuroendocrine tumors, and spindle-cell neoplasms of the upper GI tract each demonstrate a characteristic natural history. The therapeutic approach is determined by numerous factors. These factors include tissue of origin, histology, anatomic location, size, and anatomic stage. It is important to consider precursor conditions (e.g., Barrett’s esophagus, intestinal metaplasia) in the approach to surveillance, diagnosis, and ultimately therapy of upper GI malignancies.

Barrett’s esophagus is a premalignant condition characterized by specialized intestinal metaplasia (SIM) within the distal esophagus. It is considered a precursor lesion in the development of esophageal adenocarcinoma and is related to numerous factors, including chronic reflux of gastric acid and bile salts into the lower esophagus [7,8]. Similarly, intestinal metaplasia within the stomach can lead to the development of intestinal-type gastric adenocarcinoma over time. Risk factors include chronic bile reflux, high dietary salt intake, smoking, alcohol consumption, and consumption of nitrates or smoked foods [9]. *H. pylori* gastritis is also an important risk factor in the development of gastric intestinal metaplasia, with some studies showing that individuals with associated *H. pylori* infection have a six-fold increase in developing gastric cancer [10,11].

### 1.4. Approaches to Diagnosis, Staging, and Therapy

Endoscopy with high-definition white light imaging is recommended and is the standard for detecting and documenting the presence of mucosal or submucosal lesions, as shown in Figure 1. Non-invasive approaches to screening for premalignant lesions of the esophagus include devices such as the “EsophaCap” [12] and cytosponge -TFF3 that detect genetic and epigenetic alterations on samples gathered non-invasively and can be used by primary care physicians to screen for Barrett’s esophagus [13]. Non-invasive diagnostic modalities, using molecular biomarkers from a variety of body fluids to diagnose early gastric cancer, have been developed. These include “liquid-based biopsy,” which uses circulating nucleic acids. This is an exciting area of research that may change the diagnostic landscape [14].

Early endoscopic detection of premalignant lesions such as Barrett’s esophagus, gastric intestinal metaplasia, and mucosal cancer using optical chromoendoscopy techniques such as narrow-band imaging (NBI) combined with high-definition white light imaging (HD-WLE) is becoming routine across the world. Targeted biopsies from endoscopically suspicious areas or using the updated Sydney protocol are important for successful diagnosis [15,16]. In the presence of dysplasia with visible lesions or carcinoma, endoscopic ultrasound (EUS), a minimally invasive procedure, has a well-established role in the early diagnosis and locoregional staging of non-metastatic foregut malignancies [17,18,19], see Figure 2.

### 1.5. Endoscopic Approaches to Therapy

Factors including lesion size, histological features, anatomical stage, and esophageal location are important variables for determining favorability of endoscopic therapy, demonstrating curative resection in some studies [20]. Adenocarcinoma of the esophagus, limited to the mucosa (T1a)—see Figure 3—is amenable to endoscopic resection, with excellent long-term outcomes. Other tumors such as neuroendocrine tumors of the foregut may also be amenable to endoscopic therapy, including endoscopic mucosal resection (EMR) or endoscopic submucosal dissection (ESD). Specifically, these therapeutic approaches are used for resection of mucosal-based lesions, depending on the size, location, submucosal involvement, and complexity.

### 1.6. Role of Multidisciplinary Care

Treatment of upper gastrointestinal malignancies must be individualized and based on an interdisciplinary discussion in a tumor board setting, facilitated by National Comprehensive Cancer Network (NCCN) guidelines and institutional clinical pathways. In a Dutch study, the diagnostic or treatment plan was altered after a multidisciplinary tumor board discussion in over one third of cases [21]. A recent systematic review has endorsed the importance and impact of a multidisciplinary tumor board discussion in the diagnosis and management of patients with GI malignancies [22].

In addition to patient and clinical factors (e.g., functional status, comorbidities, and patient preferences), the therapeutic strategy is highly dependent on histological parameters (e.g., type of neoplasm, grade/differentiation, mitotic rate/Ki 67%, and other unique histological markers of response to certain therapies such as HER2/neu expression), anatomy/location, and, most importantly, the stage of the tumor [23,24]. The management approach may therefore involve a combination of chemotherapy, radiotherapy, endoscopic therapy, and surgery.

## 2. Esophageal Malignancies

### 2.1. Epidemiology/Pathophysiology

(i) Esophageal squamous cell carcinoma (SCC) arises proximal to the squamocolumnar junction and its pathogenesis is multifactorial in nature. It results from inflammation or other carcinogenic or mutagenic factors leading to dysplasia in situ and eventual malignant transformation. Alcohol, tobacco, caustic strictures, tylosis, thoracic radiation, and achalasia are important risk factors [25]. Dietary exposures including betel nuts and hot beverages should also be considered [26]. Areas with the highest global incidence of esophageal SCC are East Asia and Central Asia, followed by areas in Africa along the Great Rift Valley and in Uruguay in South America [27]. In the United States, Black men and women were seen to have an incidence of esophageal SCC twice that of white men and women. Between the years 1977 and 2005, SCC accounted for 87% of esophageal cancers in Black patients and 43% of white patients [28].

(ii) Esophageal adenocarcinoma (EAC) is more common in the distal and mid-esophagus and is thought to develop from specialized intestinal epithelium as a result of chronic exposure to bile, pancreatic juice, pepsin, and gastric acid. This may lead squamous cells to transform into specialized intestinal epithelium and possible dysplasia and malignancy. EAC is the predominant histological subtype of esophageal cancer in Europe and North America, with a global incidence of 0.7 cases per 100,000. Esophageal AC has surpassed SCC in a number of Western countries, including the United Kingdom, Ireland, Denmark, New Zealand, Canada, and the United States. In the United States, there are notable ethnic variations in AC incidence, with rates being higher in non-Hispanic white individuals, followed by Hispanic white, Native American, and Black American individuals, and lowest in Asian Pacific Islanders [29].

(iii) Barrett’s esophagus is a premalignant condition of the tubular esophagus, associated with gastroesophageal reflux disease, characterized endoscopically by abnormal salmon-colored mucosa located at least 1 cm proximal to the gastroesophageal junction (GEJ) and confirmed histologically as specialized intestinal metaplasia [30]. Such findings are suspicious for metaplastic change of the typical squamous epithelial lining of the esophagus to specialized intestinal epithelium with goblet cells called specialized intestinal metaplasia (SIM). Risk factors for the development of Barrett’s esophagus include male gender, Caucasian ethnicity, smoking, central obesity, increasing age, and chronic heartburn symptoms [31]. Barrett’s esophagus associated dysplasia is regarded as a precursor lesion to esophageal adenocarcinoma [8]. In patients with SIM, the risk of cancer is significantly elevated under the following circumstances: patients with specialized intestinal epithelium at index biopsy versus those without (0.38% per year vs. 0.07% per year; hazard ratio (HR) = 3.54, 95% CI = 2.09 to 6.00, *p* < 0.001), in men compared with women (0.28% per year vs. 0.13% per year; HR = 2.11, 95% CI = 1.41 to 3.16, *p* < 0.001), and in patients with low-grade dysplasia compared with no dysplasia (1.40% per year vs 0.17% per year; HR = 5.67, 95% CI = 3.77 to 8.53, *p* < 0.001) [32]. The risk of progression to adenocarcinoma may be as high as 6% per year when high-grade dysplasia is seen [33,34]. Endoscopic surveillance of Barrett’s esophagus is a powerful tool for detecting low- or high-grade dysplasia for ablative therapies or T1a adenocarcinoma which may be amenable to endoscopic resection [8,35]. High-definition white light endoscopy (HD-WLE) examination using the Seattle protocol for biopsies is the standard of care and has the same detection rate for Barrett’s intestinal metaplasia as optical chromoendoscopy techniques such as narrow-band imaging (NBI). NBI-targeted biopsies detected more areas with dysplasia, required fewer biopsies, and were therefore cost-effective [18,36]. A newer advanced endoscopic imaging modality, confocal laser endomicroscopy (CLE), has demonstrated higher sensitivity but a lower specificity when compared to NBI for detection of Barrett’s esophagus and related dysplasia in a meta-analysis of five studies [37].

### 2.2. Distant and Locoregional Staging

Staging esophageal malignancies is a crucial practice that provides prognostic information and stratifies patients to appropriate treatment modalities including endoscopic resection, surgery, radiation therapy, chemotherapy, or a combination thereof [38]. Once a lesion is identified, cross-sectional imaging with contrast-enhanced CT imaging or PET imaging is important to evaluate for metastatic disease.

Endoscopic ultrasound is a useful modality to assess the depth of invasion (T stage) of mucosa, the layer of origin of submucosal tumors and regional nodal involvement (N stage) of foregut tumors, and in some cases metastatic adenopathy. EUS is a minimally invasive procedure that allows for high-resolution imaging of the esophageal wall layers and has a diagnostic accuracy of over 80% [39]. In a meta-analysis, EUS had a high pooled sensitivity to diagnose T2 stage esophageal cancer (81.4%), T3 stage cancer (91.4%), and T4 stage cancer (92.4). EUS has a lower sensitivity in diagnosing T1 stage esophageal cancer, with a pooled sensitivity of 81.6% [38]. A recent meta-analysis looking at the ability of EUS in differentiating mucosal from submucosal cancers concluded that EUS has good accuracy (area under the curve ≥0.93) in staging superficial esophageal cancers [40], see Figure 4.

Another meta-analysis of 44 studies on EUS for staging squamous esophageal cancer showed that the overall accuracy of EUS for T staging was 79%, and for N staging, 71% [41]. In terms of N staging, EUS has acceptable accuracy, with a pooled sensitivity for diagnosing esophageal cancer of 85% and EUS fine needle aspiration (FNA) increasing the sensitivity for diagnosing esophageal cancer to 97% (95% CI: 0.92–0.99) [38].

For patients with invasive disease, imaging for staging should include PET/CT as 15% of patients present with occult metastatic disease after conventional staging with EUS or CT imaging and improved specificity of regional lymph nodes [42]. The role of PET imaging in adaptive treatment planning and assessment of tumor response is still investigational and only preliminary results have been published [43].

## 3. Treatment

### 3.1. Endoscopic Therapeutic Options for Barrett’s Esophagus Associated Dysplasia

Therapeutic options are based upon the stage of the disease at diagnosis.

Barrett’s esophagus with low- or high-grade dysplasia (HGD) without visible lesions can be treated with radio-frequency ablation (RFA) or cryoablation therapies [34,44]. A randomized, controlled trial demonstrated that (RFA) for Barrett’s esophagus with HGD was superior to surveillance with respect to the outcome of progression to esophageal adenocarcinoma but did not find any difference in progression to cancer among patients with low-grade dysplasia (LGD) [34]. A European, multicenter, randomized, controlled trial in a large group of patients with Barrett’s esophagus and LGD proved that RFA prevents progression to cancer in patients with LGD [45]. After RFA for BE dysplasia, there is >30% chance of recurrence within 5 years, but most recurrences are responsive to further endoscopic therapy [46].

Visible, discrete mucosal lesions within the Barrett’s segment, including high-grade dysplasia, T1a adenocarcinoma, and T1b adenocarcinoma with favorable histological features [47], can be approached with endoscopic resection techniques. EMR is a widely utilized endoscopic resection technique, performed using either the “Cap-assisted EMR (Cap-EMR)” technique or multiband mucosectomy (MBM) techniques with the primary goal of assessing the depth of invasion (pathological T-stage) of a visible lesion that is suspicious for intramucosal adenocarcinoma or high-grade dysplasia; see Figure 5. EMR and ESD are performed with intent of “staging” but can indeed be curative if the resection margins are “negative” or free of cancer, but the primary goal in Barrett’s esophagus is to obtain an accurate pathological T-stage of the lesion. This concept of “Staging-EMR” is demonstrated by studies that have shown a change in pathological diagnosis and/or T-staging noted in 20–30% of patients, both upstaging and downstaging, based on the pathology from the EMR specimen [48], see Figure 6.

### 3.2. Endoscopic Resection Techniques for Early-Stage Esophageal Cancer (T1a and T1b)

Endoscopic resection techniques applied to T1a and T1b mucosal cancer with favorable histological features include EMR or ESD. As mentioned earlier, EMR is primarily a “staging procedure” that provides the most accurate T-stage for mucosal lesions that are likely T1 based on EUS and/or optical chromoendoscopy [49,50]. EMR is a technique that involves submucosal injection of saline or a colloidal solution mixed with a dye such as methylene blue to create a “submucosal cushion”, followed by resection using a snare, and may be considered for lesions less than 20 mm. Techniques such as Cap-EMR, MBM, and ESD may be employed depending on anatomical location [51,52].

Cap-assisted EMR involves submucosal injection to lift the target lesion from the mucosa, isolating the lesion within a transparent “cap” mounted on the tip of the endoscope, gentle suction, followed by resection of the lesion using the pre-deployed snare within the cap using electrocautery [53]. MBM involves suction of target tissue into an endoscopic band-ligation device and deployment of a band over the target lesion followed by resection of the tissue with electrocautery snaring [54]. EMR can be either stepwise/complete (sEMR) or focal (fEMR), with focal EMR often combined with RFA. Piecemeal EMR has a high local recurrence and therefore is combined with RFA to ablate the residual BE epithelium [55]. In theory, ESD is an en-bloc resection technique with the goal being an R0 resection and is therefore a more appealing endoscopic therapeutic option.

ESD is a more complex technique that may be considered for lesions less than 20 mm. Larger lesions may be amenable to this technique depending on availability of advanced expertise and resources [56]. Specifically, this technique involves the creation of a submucosal tract to dissect lesions restricted to the mucosa (T1a or selected T1b), for complete resection. Factors including lesion size less than approximately 20 mm, anatomical stage (T1a), and middle thoracic and lower thoracic esophageal locations are favorable variables for successful ESD, demonstrating curative resection in some studies [20,56]. Additionally, T1a m2, T1a m3, and select T1b lesions with favorable histological features [57] are amenable to ESD, with comparable outcomes to esophagectomy [58,59]. For these types of lesions, current evidence shows that EMR and ESD are comparable in terms of complication rates, referral to surgery, positive margins, lymph node positivity, local recurrence, and metachronous cancer. When compared to piecemeal EMR resection, ESD may offer some advantages, but data are limited [60,61]. Meta-analyses comparing both of these therapies revealed comparable outcomes in terms of Barrett’s eradication, low rates of recurrence, and have shown non-inferiority of EMR to ESD for therapy of Barrett’s related cancer or GEJ neoplasia, with similar complication rates [15,62].

### 3.3. Multimodality Therapy for Locally Advanced Esophageal Cancer

For locally advanced tumors (T2 or with nodal involvement), a combination of neoadjuvant chemoradiation therapy and surgical therapy can be used, with systemic chemotherapy being reserved for metastatic disease (Stage IV) with possible consolidation chemoradiation to sites of disease involvement if the patient responds well [63].

Based on current guidelines, esophagectomy for operable non-metastatic patients with T1b or greater primary lesions and/or any nodal disease should be performed with one of several techniques including: transhiatal, transthoracic, three field, and, increasingly, minimally invasive approaches [64]. Meta-analyses of minimally invasive esophagectomy, which includes laparoscopic/thoracoscopic and robotic approaches, have shown no difference in survival and improved or no difference in complications. Robotic-assisted esophagectomy is becoming increasingly popular, with demonstrated reduction in cardiac and pulmonary complications when compared with open esophagectomy. Technologic advances in the minimally invasive and robotic platforms are also rapidly developing and may allow for further innovation with these techniques [65,66]. While surgery alone can be considered for early-stage, low-risk adenocarcinoma (<3 cm and well differentiated) and early-stage squamous cell carcinoma, trimodality therapy with neoadjuvant chemoradiation followed by surgery is now a preferred treatment pathway for more advanced disease and requires multidisciplinary management [67]. Less commonly, postoperative chemoradiation for pathologic T3 and/or node positive disease may be indicated if a GEJ cancer was treated with preoperative chemotherapy, while palliative radiotherapy may be indicated for treatment of significant dysphagia or bleeding. In definitive treatment of inoperable patients or those who decline esophagectomy, concurrent chemoradiation should be prioritized, followed by imaging and endoscopic surveillance to determine clinical complete response [68]. While clinical outcomes and morbidity for these patients are still suboptimal, there are a growing number of long-term survivors after definitive chemoradiation and investigations are being made into radiotherapy dose escalation, proton therapy, and inclusion of targeted therapy [69,70,71,72].

Three-dimensional conformal planning for radiation therapy can be utilized; however, more institutions have implemented intensity-modulated radiation therapy (IMRT) to reduce cardiac and lung dosing, which allows for a simultaneous integrated boost (SIB) or sequential boost techniques [73,74,75]. This approach is important as cardiac dose is increasingly seen as an independent risk factor for reduced survival [76] and has been correlated with excess G3+ cardiac toxicity [77,78].

Fiducial markers can be placed endoscopically to delineate the extent of disease with high technical success with a small risk of migration [79,80,81]. Stable fiducial markers improve the reliability of target volume delineation and assessment of respiratory tumor motion with four-dimensional CT (4D-CT) simulation as a direct visual correlate of tumor extent. Combined with fused PET/CT, fiducials reduce the margins for treatment planning due to improved confidence of accurately defined gross tumor volumes (GTVs) [82] and facilitate daily image-guided radiation therapy (IGRT) during treatment [83], see Figure 7 and Figure 8. Moreover, 4D-CT imaging at simulation has a greater benefit in GEJ and gastric tumors due to the propensity of respiratory motion [84] and aids in internal target volume construction and planning target volume expansions during treatment planning. Radiation treatments generally are conventionally fractionated (1.8–2.0 Gy daily); however, previously mentioned SIB techniques treating at 2.2–2.25 Gy daily can push dose to gross disease past 60 Gy. If given sequential neoadjuvant chemotherapy, the extent of original disease is often included in the clinical target volume receiving 4500–5040 cGy and boost is only directed to residual gross disease. Clinical target volumes extend 3–4 cm craniocaudal and 1 cm radially and are edited off anatomical structures that are clinically uninvolved such as vertebral bodies, trachea, aorta, lung, and pericardium. For middle and upper thoracic tumors, the at-risk periesophageal and adjacent mediastinal lymph nodes and, for distal esophageal and GEJ tumors, the celiac lymph nodes are covered. Multi-institutional consensus contouring guidelines for IMRT in esophageal and gastroesophageal tumors have been published [85].

### 3.4. Neoadjuvant Radiation Therapy Alone and Adjuvant Therapy

Neoadjuvant treatment carries the theoretical gains of tumor downstaging, margin attenuation, and improved control of regional disease. However, limited benefit has been observed across several randomized trials investigating the role of radiotherapy alone in preoperative or adjuvant treatment of esophageal cancer. Early studies on patients who received 4000 cGy followed by surgery 1–4 weeks after found no difference in resectability or survival compared to surgery alone [86,87]. A trial from the European Organization for Research and Treatment of Cancer (EORTC) also demonstrated no difference in survival, but noted a lower rate of local failure with the addition of radiotherapy to 46% from 67% [88]. More recent studies that reported improved survival included patients that received chemotherapy but had an imbalance of lower stage tumors conflicting the interpretation of the results [89]. These findings have been confirmed in meta-analyses that demonstrate no statistical difference in survival, or at best, the suggestion of a very modest clinical benefit of <4% with neoadjuvant radiotherapy alone [90,91].

Recommendations for postoperative or adjuvant treatment are generally driven by adverse pathological features such as positive lymph nodes, positive margins, or locally advanced disease found at time of surgery. While the goal of local therapy is to reduce the risk of local recurrence, often, toxicity is greater due to larger treatment volumes, efficacy is reduced due to a hypoxic tumor bed, and dose is limited by the surgical anastomosis. In two historic series, adjuvant radiotherapy alone showed no survival benefit compared to observation [92,93]. However, similar to preoperative treatment, there are trends towards improved local control after delivery of 4500–5500 cGy [93] after subtotal resection [92]. Retrospective data and population-level analyses show a benefit of adjuvant chemoradiation in pathologically node positive [94] but not node negative disease [95]. The benefit of the addition of chemotherapy to adjuvant radiation is currently being investigated in a phase II/III protocol seeking to add paclitaxel and cisplatin/nedaplatin to 50.4–54.0 Gy of radiotherapy for pathological stage IIB-III disease [96].

### 3.5. Preoperative Chemoradiation

For locally advanced resectable disease, the CROSS trial (Chemoradiotherapy for Oesophageal Cancer Followed by Surgery Study) established a new paradigm towards total neoadjuvant treatment with chemoradiation followed by surgical resection in esophageal cancer [97]. Other comparisons to neoadjuvant chemotherapy alone in the German POET trial [98] and the NeoRes trial [99] demonstrated improvements, with pathological downstaging with the addition of 30 and 40 Gy of chemoradiation, but did not translate to statistically significant survival differences, likely limited by increased toxicity in the experimental arms with concurrent cisplatin/flurouracil. Within CROSS, radiation therapy was delivered to 4140 in 1.8 daily fractions with concurrent paclitaxel/carboplatin and was compared to surgery alone. Neoadjuvant chemoradiation was associated with a 25-month improvement in median overall survival (49 vs. 24 months) and was notable for a 47% 5-year survival rate and 29% overall pCR rate that was significantly different between histologies (28% adenocarcinoma vs. 49% squamous) [100]. Additionally, preoperative chemoradiation reduced hematogenous metastases (35% vs. 29%) and peritoneal carcinomatosis (14% vs. 4%). There was no difference in operative mortality between the arms or long-term quality of life, with all endpoints declining after surgery and subsequently improved and stabilized between 6 and 12 months afterwards [101].

The CROSS trial excluded patients with tumor diameter greater than 8 cm and patients who had lost more than 10% of their original body weight. In these patients, upfront chemotherapy alone can be considered as they are at high risk of clinical decline during chemoradiation. Therapeutic response to chemotherapy may enable a reduction in irradiation volumes to mitigate pulmonary toxicity with concurrent taxols or motivate one to switch to an alternative regimen before or during chemoradiation. This strategy may also be considered in patients with advanced primary tumor (T4) or regional nodal burden (bulky nodes or cN2+ disease) as they are at higher risk of distant progression and theoretically may derive greater benefit from intensification of systemic therapy. The outcomes from the CROSS trial remain some of the best to date with this strategy, and similar patterns were noted in the Chinese multicenter NEOCRTEC5010 trial that demonstrated improved median survival (66 vs. 100 months), disease-free survival, and R0 resection rate with neoadjuvant vinorelbine/cisplatin with 40 Gy compared to surgery alone [102]. Future strategies seek to compare FOLFOX to paclitaxel-carboplatin with concurrent neoadjuvant radiation for both squamous and adenocarcinoma in the PROTECT-1402 trial (Preoperative Chemoradiation for Resectable Esophageal and Junctional Cancer) [103]. Given the advances in surgical and radiotherapy techniques and corresponding declines in perioperative morbidity and mortality, trimodality therapy is now a preferred approach.

### 3.6. Definitive Radiation and Chemoradiation

Historically, definitively treated patients generally carry significant comorbidities precluding surgery or are opposed to the morbidity of resection. Alternatively, the surgical options for patients diagnosed with cervical esophageal cancer are often limited and these patients are treated within the paradigm of head and neck cancer with definitive chemoradiation. The use of radiation alone in the modern era is generally considered to be palliative, with 5-year survival of <10% [104,105], while the importance of concurrent chemotherapy has been demonstrated historically by RTOG 8501 where, at 5 years, no survival was noted with dose escalated 64 Gy radiotherapy alone versus 27% survival with 50 Gy and concurrent four cycles of fluorouracil and cisplatin [106]. Initial forays of radiotherapy dose escalation in the definitive treatment of esophageal cancer showed disappointing clinical results. INT 0123 investigated chemoradiation with cisplatin/fluorouracil and compared 5040 cGy to dose escalation to 6480 cGy [107]. A total of 236 patients were enrolled with T1-4, N0-1 disease with 85% squamous cell carcinoma. Overall, the target volumes for this trial were relatively large, with 5-cm superior/inferior and 2-cm radial esophageal expansion as well as a 2-cm isometric expansion from tumor for the boost to gross disease. The dose escalation arm experienced a 10% treatment-related G5 toxicity rate; however, 64% of the mortality occurred prior to surpassing the dose of the control arm, potentially implicating the toxic combination of cisplatin and fluorouracil. Median survival approximated 15 months; however, 2-year locoregional failure was greater than 50%. RTOG 9207 was a phase I/II study that utilized the same control arm as INT 0123 but dose-escalated through the use of predominantly high dose rate (HDR) brachytherapy with iridium-192 to 15 Gy in three weekly fractions [108]. Unfortunately, toxicity from this strategy was high, with 12% fistula rates, 24% G4 toxicity, and 10% treatment-related deaths. Subsequently, the brachytherapy dose was dropped to 10 Gy in two weekly fractions and no fistulas were noted in the 10 patients treated with the intermediate dose. However, overall clinical outcomes remained poor, with 1-year survival of 50% and 37% local control.

With a goal of improving on the near 50% local failure rate seen in previous studies as well as overcoming high rates of persistent disease [107,108,109], dose escalation in the modern era appears to be more tolerable but still carries an elevated risk of toxicity as well as minimal evidence of clinical benefit. In preliminary results, both a sequential boost to 60 Gy with cisplatin/docetaxel [110] and simultaneous integrated boost to 61.2 Gy with carboplatin/paclitaxel [111] failed to show improvement in overall or progression free survival, despite the latter regimen carrying a 13% rate of G4 toxicity and 10% treatment-related toxicity in the high dose arm. MD Anderson has published results with their SIB technique of a high dose gross tumor volume (GTV) plus 3 mm and a planning target volume (PTV) margin of 5 mm with concurrent chemotherapy, demonstrating encouraging 66% local control at 2 years, acute G3 toxicity in 23%, and only stricture-related late G3 toxicity in 7% [112].

Newer strategies include proton therapy, which carries dosimetric benefits and higher relative biological effectiveness than photons due to the physical nature of the particle. Early retrospective data have not demonstrated an oncological benefit in comparison to photon therapy [113]; however, there is some evidence of a decrease in treatment-related toxicity which, in larger prospective studies, may carry clinical significance [114]. More work remains to further elucidate the benefits from these techniques.

### 3.7. Therapy for Metastatic Esophageal Cancer

Palliative radiotherapy is commonly delivered for esophageal obstruction in metastatic patients as a significant improvement in patient quality of life. Other indications are severe pain, chronic blood loss, or nausea due to tumor mass effect. While external beam radiotherapy and self-expanding metal stents (SEMS) to palliate dysphagia remain more common palliative techniques, growing literature supports the safe use of intraluminal brachytherapy for durable palliation with caution for fistulation or stenosis [115].

A meta-analysis of 53 studies (mostly RCTs) on palliation of dysphagia in inoperable esophageal cancer concluded that SEMS insertion is safe, effective, and quicker in palliating dysphagia compared to other modalities. The authors added that, “Brachytherapy might be a suitable alternative to SEMS in providing a survival advantage and possibly a better quality of life” [116].

The goal of chemotherapy in patients with stage IV metastatic esophageal cancer is to improve survival and quality of life; several chemotherapeutic agents have been tested and used in the past several decades and proven to be effective in achieving this goal. Unfortunately, survival has remained poor and rarely surpasses one year.

Combination of 5 fluorouracil (5 fu) and platinum agents is an acceptable first-line treatment option and is considered the standard of care in this setting; other regimens including paclitaxel with platinum regimen, irinotecan plus 5-fu are recommended as well. Three drug combinations such as modified DCF (Docetaxel, Cisplatin, 5 fu) are usually given to patients with high volume disease, young age, and good performance status who might benefit from a higher response rate. Single-agent treatment is recommended for patients with low volume disease and or poor performance status.

Several targeted therapies have been tested as front-line and in combination without significant improvement in overall survival until the ToGA trial was conducted. The ToGA trial was a randomized phase III trial where patients with Her-2 expressing tumors were randomized to chemotherapy with or without trastuzumab (Her-2 monoclonal antibody); patients who received the combination had better overall survival, leading to the approval of this combination in this patient population [117].

Recently, at the European Society of Medical oncology meeting in 2020 (ESMO 2020), three important trials were presented incorporating immune checkpoint inhibitors in the front-line treatment of esophageal and gastric cancer. The CheckMate 649 trial evaluated nivolumab plus chemotherapy versus chemotherapy alone as first-line treatment in patients with non-HER-2-positive advanced gastric and GE junction adenocarcinoma. The addition of Nivolumab improved overall survival and progression-free survival in patients with PD-L1 combined positive score (CPS) ≥5 tumors. Improvements were also observed in patients with PD-L1 CPS ≥1 tumors and in the overall patient population. In another study, the ATTRACTION 4 trial, which was performed only in Asian patients and the primary endpoints were designed for all-comers, rather than a specific CPS value, first-line treatment with nivolumab plus chemotherapy improved the co-primary progression-free survival endpoint, but not overall survival [118].

The third trial, presented at the same meeting, the KEYNOTE 590 trial, examined first-line chemotherapy with or without pembrolizumab in patients with squamous cell carcinoma of the esophagus, adenocarcinoma of the esophagus, or Siewert-type GE junction adenocarcinoma. It demonstrated that pembrolizumab plus chemotherapy improved overall survival in patients with squamous cell carcinoma of the esophagus with PD-L1 CPS ≥10 tumors, all squamous cell carcinomas, all patients with CPS ≥10, and the study population as a whole. Progression-free survival was also improved. It is expected that these trials will change the landscape of the treatment of esophageal cancer worldwide [118].

Several chemotherapy and targeted agents have been studied in the second-line treatment of esophageal and gastric cancer and beyond. Chemotherapy improves survival compared to placebo, single-agent paclitaxel with or without ramucirumab, a monoclonal antibody targeting the VEGF pathway, which is a commonly used second-line regimen with overall survival approaching 9 months. Single-agent ramucirumab, irinotecan, and immune checkpoint inhibitors are all approved for the second-line treatment as well, with modest benefits. Patients with microsatellite instable disease should be treated with immune check point inhibitors at any point during the course of their disease [118].

Several ongoing trials are testing other targeted therapies and are beyond the scope of this review.

Patients with inoperable esophageal cancer and with high-grade dysphagia were randomized to receive a self-expandable metal stent (SEMS) alone (Group I), versus a combination of SEMS followed by external beam radiation (over 2 weeks) (Group II). Dysphagia scores improved significantly in both groups following stent insertion. However, dysphagia relief was more sustained in Group II than in Group I (7 vs. 3 months, *p* = 0.002), and overall median survival was significantly higher in Group II than in Group I [119]. A recent meta-analysis of several RCTs evaluating the efficacy of SEMS alone vs. SEMS combined with radiotherapy concluded that the combination of SEMS and radiation significantly improves the overall survival as well as leading to improvements in quality of life scores [120].

In summary, the current trend in the literature shows that the best oncological outcomes are associated with trimodality therapy with neoadjuvant chemoradiation. This has been supported by recent meta-analyses that conclude that compared with neoadjuvant chemotherapy, neoadjuvant chemoradiotherapy should be recommended, with a significant long-term survival benefit in patients with cancer of the esophagus or the GEJ [121,122]. While treatments and outcomes are improving, a large proportion of patients fail locally, and addition of biologically targeted agents or local therapy intensification may prove to be beneficial.

For patients who are considered to be surgical candidates, neoadjuvant chemotherapy with weekly carboplatin and paclitaxel in combination with radiation followed by surgical resection is the standard of care. This treatment is based on the CROSS trial, where neoadjuvant therapy resulted in improved overall survival compared to esophagectomy alone [97]. Definitive chemotherapy and radiation is a reasonable option for patients who are not surgical candidates or with cervical/mid-esophageal SCC. On rare occasions, patients who are thought to have T1N0 disease are upstaged during esophagectomy; the role of adjuvant therapy in this setting is not clear and carries significant toxicity. The addition of Her-2 targeted therapy with trastuzumab (Her-2 antibody) to the neoadjuvant therapy did not result in improved outcome [123]. Several ongoing trials are currently evaluating the addition of immune checkpoint inhibitors in both the neoadjuvant and the adjuvant settings. These results are highly anticipated.

As previously described, neoadjuvant radiation therapy in addition to chemotherapy plays an important role in the multimodality treatment of locally advanced esophageal cancer. Fiducial markers have been integrated into the management of multiple malignancies to guide more precise delivery of radiation therapy (RT).

## 4. Gastric Malignancies

There are several gastric tumors, but we will focus our discussion on gastric adenocarcinoma, gastric GIST, NET, and MALT lymphoma.

### 4.1. Gastric Adenocarcinoma

Gastric adenocarcinoma, commonly referred to as gastric cancer (GC), represents the second-most common cancer worldwide and the fifth leading cause of cancer deaths. Men are observed to have approximately double the incidence of gastric cancer as women [124]. Globally, almost two thirds of cases are observed in East Asia, with China representing 43% of cases annually [2]. Gastric adenocarcinoma is histologically divided into intestinal and diffuse types. The intestinal type is distinguished by gland-like structures, whereas the diffuse type is characterized by poorly differentiated cells. Gastric adenocarcinomas arise as a result of a complex multifactorial process. Environmental factors include dietary exposures to sodium, nitrates, nitrites, nitrosamines, and lack of refrigeration. *H. pylori* infection and its associated proinflammatory milieu also represent important risk factors. There are multiple virulent strains of *H. pylori*; however, they are divided into two major subpopulations based on their ability to produce a 120–145 kDa immunodominant protein called cytotoxin-associated gene A (*CagA*) antigen. Strains that express the (*cagA*) pathogenicity island (PAI), a 40-kilobase DNA segment, are associated with increased inflammatory response and important clinical outcomes including peptic ulcers and gastric cancer [125]. Approximately 60% of *H. pylori* strains isolated in Western countries express *cagA* PAI whereas nearly all *H. pylori* from East Asian isolates express *cagA* PAI [126]. Additional predisposing conditions for gastric cancer include intestinal metaplasia and pernicious anemia. Rarely, genetic mutations such as CDH-1 contribute to familial predisposition to gastric adenocarcinoma [2].

### 4.2. Early Gastric Cancer (EGC)

Early Gastric Cancer: Endoscopic Diagnosis and Therapy

High-definition white light endoscopy (HD-WLE) combined with optical chromoendoscopy techniques such as narrow-band imaging (NBI) and targeted biopsies is the standard for early diagnosis of gastric cancer; see Figure 9. NBI increases the diagnostic yield for advanced gastric premalignant lesions compared to white light endoscopy [127]. The updated Sydney protocol is a standardized biopsy technique recommended for mapping the stomach to screen for atrophy and gastric intestinal metaplasia [128].

According to the Japanese Gastric Cancer Association, nonulcerated EGCs confined to the mucosa (T1a), with differentiated histology, and ≤20 mm, have low risk for lymph node metastasis and therefore are appropriate for endoscopic resection using the ESD technique. The criteria for ESD were expanded to include (1) nonulcerated differentiated EGCs of any size, (2) ulcerated differentiated EGCs <30 mm, or (3) differentiated EGCs <30 mm with superficial submucosal invasion (SM1; <500 μm below the muscularis mucosae) [129]; see Figure 10.

Asian endoscopists have applied the expanded criteria for EGC and have demonstrated that ESD has high curative rates comparable with those of traditional gastrectomy [113]. ESD is the standard of care for the management of EGC, meeting the expanded criteria as outlined by the Japanese guidelines [130]. The expanded Japanese criteria for resection were applied in the European setting with an en bloc resection rate of 98.4% for the standard guideline criteria and 89.0% for expanded criteria, and the R0 resection rate was 90.2% and 73.6%, respectively. Based on the excellent long-term outcome using the expanded criteria in EGCs in Western countries, ESD was recommended as the treatment of choice for intramucosal nonulcerated EGCs regardless of their diameter [131].

However, the applicability of the current Japanese guidelines for endoscopic resection of EGC outside of Asia has been questioned based on differences in tumor characteristics and survival rates between Asian and Western populations [132,133]. The majority of undifferentiated intramucosal early gastric cancer (EGC) cases do not have lymph node metastasis but endoscopic resection has not been accepted as an alternative treatment to surgery for this subgroup of EGC.

Abe et al. have identified two independent risk factors for lymph node metastases: a tumor ≥20 mm and presence of lymphatic involvement on endoscopic resection specimens. They recommended that undifferentiated EGC <10 mm without lymphatic involvement may be treated by endoscopic resection primarily [134]. If lymphatic involvement is seen on the endoscopic resection specimen, additional surgery is warranted.

Others have suggested a size threshold of 20 mm or less in tumors without lymphatic-vascular capillary involvement or ulceration as suitable candidates for endoscopic resection based on low rate of lymphatic metastases, incidence of lymph node metastasis, and the feasibility of endoscopic resection for undifferentiated-type early gastric cancer [135].

### 4.3. Endoscopic Ultrasound in Staging of Gastric Cancer

EUS accuracy has historically been suboptimal in the preoperative staging of gastric cancer and it is recommended that EUS should be combined with other modalities for the preoperative staging of gastric cancer [136]. A meta-analysis evaluated the diagnostic accuracy of EUS for the preoperative locoregional staging of GC and the Cochrane Collaboration Group evaluated 66 articles on GC staged with EUS. EUS sensitivity and specificity to discriminate T1-T2 from T3-T4 lesions to be 86% and 90%, respectively [137]; see Figure 11. In a prospective study of patients with GC before and after neoadjuvant chemotherapy, the overall accuracy of EUS was around 80% and EUS performed better than PET-CT for N staging and restaging [138]. Similar to esophageal cancer, patients with potentially resectable GC with T2 disease and higher or any evidence of nodal involvement have significant benefits of adjuvant or neoadjuvant therapy compared to surgery alone. In patients with GC, a complete staging with laparoscopy and peritoneal washing is recommended if feasible, followed by perioperative treatment. Several regimens have been evaluated in large randomized trials for locally advanced potentially resectable gastric cancer including adjuvant chemoradiotherapy: Intergroup 0116(2) and Cancer and Leukemia Group B(3) (CALGB 80101), perioperative chemotherapy [139] and more recently the FLOT regimen ^133^, and adjuvant chemotherapy alone in Asia. All showed benefits of perioperative treatment. In the United States, the majority of patients receive perioperative chemotherapy with or without radiation. Similar to esophageal cancer, several targeted agents such as VEGFR antibodies and Her-2 antibodies have been integrated in the perioperative setting, with limited benefits. Several ongoing studies are evaluating the addition of immunotherapy to chemotherapy in this setting as well.

### 4.4. Gastrointestinal Stromal Tumor (GIST)

Gastrointestinal stromal tumors arise anywhere along the digestive tract and are thought to arise from the interstitial cells of Cajal or a common progenitor cell, based on common immunophenotypes. These are found in patients with advanced age and are most commonly found in the stomach, duodenum, jejunum, and ileum. Histologically, these may be spindle-cell, epithelioid, or mixed type. These tumors may express mutations in tyrosine kinase (TK), KIT receptor, or platelet-derived growth factor (PDGF). Histologically, these are distinguished with spindle-cell morphology. These may be localized or metastatic (peritoneum, liver). Disease progression is related to several key factors including location, size, and mitotic index [140]; see Figure 12 and Figure 13.

Surgical resection is the preferred treatment for localized GIST tumors. The goal of surgery is to achieve a negative margin resection without causing tumor rupture. For gastric GISTs, a wedge resection can be performed, avoiding formal gastrectomy. Lymphadenectomy is not indicated. Laparoscopic or minimally invasive resections have been associated with less blood loss, lower rates of complications, and shorter hospital stays without compromise of oncologic principles [141]. With some GIST tumors, either small or predominantly intraluminal ones, they may be difficult to fully visualize and anticipate margins by visualization laparoscopically. Endoscopic assistance in this circumstance can provide the added benefit of intraluminal assessment and delineation of margins; this combined endoscopic and laparoscopic approach (i.e., laparoscopic and endoscopic cooperative surgery) can help avoid more major gastric resections [142]. The goal of surgery is to achieve complete resection, and if feasible, then an upfront resection is recommended. If a highly morbid operation is required, or one requiring multivisceral resection, neoadjuvant imatinib can be considered [143]. This may allow for downsizing of the tumor, to avoid a more morbid operation [144]. A biopsy should be performed, and tumor genotype could help guide neoadjuvant treatment. Imatinib is the treatment of choice in this setting; however, tumors that have a platelet-derived growth factor receptor-alpha (*PDGFRA*) D842V mutation, or a succinate dehydrogenase (SDH)-deficient or neurofibromatosis (NF)-related GIST, are considered resistant to imatinib and will not benefit from neoadjuvant treatment; in these cases, upfront surgery is recommended. Tumors that harbor an exon 9 *KIT* mutation should be treated with an initial dose of 800 mg per day. Patients should be treated to the maximum response followed by surgical resection.

Based upon the results of the Scandinavian Sarcoma Group (SSG) XVIII adjuvant trial, daily imatinib treatment for 36 months is currently the standard of care for patients with high-risk disease [145]. Patients with metastatic disease are best treated with systemic therapy with imatinib and the other approved TKIs, with surgical debulking reserved for those who respond well and can achieve complete resection.

### 4.5. Neuroendocrine Tumors

Neuroendocrine tumors can arise throughout the digestive tract and other areas of the body. Gastric neuroendocrine tumors account for 7–8% of all neuroendocrine tumors [146]. These tumors are derived from the enterochromaffin-like cells in the stomach and are divided into three distinct types (I–III).

Type I neuroendocrine tumors are the most common and represent 70–80% of gastric neuroendocrine tumors. They can be seen in association with chronic atrophic gastritis. Due to resultant chronic achlorhydria, G-cell hyperplasia happens and results in increased gastrin secretion and hypergastrinemia. These tumors typically occur in the mucosa or submucosa and are more indolent. Smaller tumors (less than 1 cm) may be amenable to endoscopic resection (EMR and ESD) and surveillance via upper endoscopy, endoscopic ultrasound.

Type II neuroendocrine tumors are associated with Zollinger Ellison syndrome (Multiple Endocrine Neoplasia I). These are the least-frequently occurring, representing between 5% and 8% of cases. Metastatic potential is estimated at approximately 30% [147].

Type III and Type IV neuroendocrine tumors are more aggressive and may be metastatic at presentation. Therapeutic approaches may be more systemic.

The three types of gastric neuroendocrine tumors have previously been described, as well as their relationship to gastrin levels. Type II gastric neuroendocrine tumors are rare and found in the setting of increased acid production from hypergastrinemia secondary to gastrinoma. Type 2 gastric NET should be resected. Type III gastric NET are found in the setting of normal gastrin levels and exhibit a more aggressive behavior; they are treated surgically in a manner similar to adenocarcinoma: formal gastric resection and regional lymphadenectomy. Small, low-grade, Type III gastric NET without evidence of lymphovascular invasion can be treated with wedge or endoscopic resection [148].

Type I and II G-NETs appear either as small (<10 mm) polypoid lesions, or more frequently, as smooth hemispherical submucosal lesions, that may appear yellow or red in color. EUS is useful for assessing the depth of the gastric NETs and their location within the layers of the gastric wall. They usually arise from the second (deeper mucosal) or third (submucosal) echo layers and are identified as hypoechoic intramural structures [149].

EUS-FNB is a useful modality in the diagnosis of gastric NET and, in the appropriate settings, can provide adequate tissue for histological diagnosis [150]. Endoscopic resection using the band ligation or cap technique is an established technique for resection of gastric NETs <1 cm [151]. Currently, there is no role of adjuvant therapy following surgical resection in these patients per NCCN guidelines [152].

### 4.6. Gastric Lymphoma

There are two major types of primary gastric lymphoma: mucosa-associated lymphoid tissue (MALT) gastric lymphoma or diffuse large B-cell lymphoma (DLBCL) of the stomach. MALT lymphoma is a type of extra-nodal low-grade marginal-B cell lymphoma which may occur in the stomach, small bowel, and other organs. These lesions may present as erosions or nodular masses and are distinct from gastric adenocarcinoma [153]. *H. pylori* is noted as a causative factor for MALT lymphoma and eradication of *H. pylori* can result in remission in 50–90% of cases. Therefore, eradication of *H. pylori* is regarded as a first-line treatment for gastric MALT lymphoma, with remission favorably associated with early lesions, lesions limited to the mucosa, and absence of the *API2MALT1* mutation [154]. The role of endoscopy is one of diagnosis, response to therapy, and surveillance, with EUS demonstrating value in predicting submucosal and regional lymph node involvement [155].

### 4.7. Staging

Laparoscopic assessment of the peritoneal cavity for patients with advanced gastric cancer (T2+) is an established method of detecting radiographic-occult peritoneal disease, both in Eastern and Western countries. Over 30% of patients may be found to have peritoneal disease or positive peritoneal cytology [156,157]. Positive peritoneal disease or cytology renders a stage IV prognosis and diagnosis and can help temper treatment expectations. For those with Tis or T1a cancers that fit the Japanese guidelines, consideration of endoscopic resection techniques such as ESD is a reasonable approach. For distal tumors, subtotal distal gastrectomy is recommended. For proximal tumors, total gastrectomy with roux-y esophagojejunostomy is preferred. Concomitant splenectomy is no longer performed as morbidity is increased without any survival benefit [158]. For patients with localized or regional disease, total formal gastrectomy with modified D2 lymphadenectomy is recommended in the US, with the goal of examining at least 15 lymph nodes, following a multidisciplinary review for consideration of perioperative chemotherapy. In Eastern countries, modified D2 lymphadenectomy is considered standard of care. The 15-year follow-up of the Dutch D1D2 trial did not show any overall survival benefit; however, there was lower local and regional recurrence and gastric cancer-related deaths. This was not shown in other trials, thereby rendering modified D2 lymphadenectomy as a recommendation in NCCN guidelines [159,160]. Laparoscopic approaches to gastrectomy have also been investigated, mostly in Eastern countries, and have been shown to have similar oncologic outcomes to open gastrectomy [161,162].

### 4.8. Treatment

Primary gastric cancer encompasses tumors with an epicenter classification of Siewert III superiorly (>2 cm distal to GEJ) and extends inferiorly to tumors arising from the gastric antrum and pylorus. As in most upper intestinal GI malignancies, surgery by gastrectomy with lymph node dissection is considered the primary curative therapy. Similar to esophageal cancer, no adjuvant therapy is generally recommended for early-stage pT1N0 disease and these tumors may be adequately treated by endoscopic resection and close follow-up. However, if pT2+ or pN+, adjuvant chemotherapy is given with consideration for radiotherapy based on risk factors of positive margins or inadequate nodal dissection (D1, or <15 total LN). More recent evidence supports a survival benefit of neoadjuvant chemotherapy for resectable disease. In patients with locally advanced disease, neoadjuvant chemotherapy with or without chemoradiation can be considered [98]. For inoperable patients, definitive chemoradiation may be given; however, outcomes are generally poor.

Radiotherapy simulation and treatment planning for GC generally follows a similar paradigm to GEJ tumors with 4D-CT simulation and IGRT to EUS-placed fiducial markers or surgical clips. Contouring guidelines have been published to aid delineation in the setting of esophagogastrectomy, total gastrectomy, and subtotal gastrectomy [163]. Generally, dose in the adjuvant setting is conventionally fractionated to 45 Gy with a 5.4–9.0 Gy boost for microscopically positive disease or higher for gross residual disease. Nodal regions at risk are dependent on the primary site of disease as well as extent of invasion and commonly include the perigastric, celiac and celiac axis, SMA, as well as supra- and infra-pyloric nodal basins. The distal paraesophageal, porto-hepatic, and pancreaticoduodenal lymph nodes may be covered depending on primary tumor location. Elective node coverage can be omitted in patients with pT2-3N0 disease with D2 nodal dissection with ≥15 LN removed [164].

Acute toxicity during treatment may be significant with fatigue, nausea, vomiting, dyspepsia, or gastritis and may be particularly pronounced in the adjuvant setting after more extensive gastrectomies. Late toxicity including GI stricture, chronic kidney insufficiency, and secondary malignancies has been associated with gastric radiotherapy. These toxicities were particularly pronounced with two-dimensional fields as utilized in INT0116 that resulted in 33% G3+ GI toxicity and upwards of 50% hematologic G3+ toxicity [165]. Modern IMRT techniques have a dosimetric benefit in silico for tumor coverage as well as organ sparing [166,167], and early reports are favorable regarding toxicity in preoperative [168] and postoperative settings [169,170].

#### 4.8.1. Adjuvant Chemoradiation

The use of adjuvant radiotherapy in the treatment of GC has historically been limited in dose due to organ and anastomosis tolerance as well as high toxicity rates. Furthermore, as our understanding of the impact of the extent and quality of surgical resection has broadened, patient selection and risk stratification has improved. The landmark INT0116 trial demonstrated a survival benefit of adjuvant chemoradiation versus observation after gastrectomy with D0/D1 lymph node dissection [165]. This population consisted of predominantly advanced disease with nearly 70% pT3-4 and 85% pN+, and with extended follow-up of 10.3 years, chemoradiation offered a survival benefit (HR 1.32, *p* < 0.001) and nearly halved locoregional relapses [171], while more extensive D2 lymph node dissection initially resulted in higher rates of surgical morbidity and postoperative mortality than D1 [172], long-term follow-up demonstrated improvements in locoregional recurrence and gastric cancer specific mortality [160]. Within this domain, the data to date mostly show no benefit in the addition of radiation to adjuvant chemotherapy due to the outcomes of the CRITICS [173] and ARTIST trials [174,175].

The ARTIST I trial enrolled 458 patients with pathologic stage IB-IV (M0) disease who underwent an R0 gastrectomy and D2 lymph node dissection to adjuvant capecitabine and cisplatin (XP) for six cycles alone vs. chemoradiation to 45 Gy with concurrent capecitabine sandwiched by two cycles of XP [174]. With extended follow-up, there was no significant difference in OS and DFS; however, on subgroup analysis, radiotherapy was associated with improvement in DFS in patients with pN+, high lymph node ratio and intestinal-type histology [175]. These findings were explored in ARTIST II in pathologically node positive patients; however, no benefit was noted with chemoradiation against combination chemotherapy alone (HR 0.91, *p* = 0.667) [176]. The trial was terminated early but final publication is anticipated.

Finally, CRITICS included 788 non-metastatic stage IB-IVA gastric cancer patients who were randomized upfront to adjuvant chemoradiation or chemotherapy alone [173]. All patients were given neoadjuvant chemotherapy consisting of epirubicin, capecitabine, cisplatin/oxaliplatin (ECF/ECX) for three cycles followed by gross total resection, with 79% D1+ resections and a median of 20 lymph nodes removed. After surgery, patients received either three cycles of chemotherapy or chemoradiation to 45 Gy with weekly cisplatin and capecitabine. Unfortunately, the dropout rate during adjuvant therapy was high in both arms, approximating 50%, limiting conclusions of the study, but median survival for adjuvant chemoradiotherapy was numerically less at 37 months vs. 43 months with adjuvant chemotherapy but not statistically different. Perioperative morbidity was high, with a pooled 66% G3+ toxicity rate with preoperative chemotherapy and similar G3+ GI toxicity rates near 40% between both adjuvant treatments, highlighting the toxicity of these regimens.

The authors of CRITICS concluded that in the era of delivery of intensive neoadjuvant therapy, the impact of adjuvant chemotherapy or chemoradiation may be minimalized, especially as rates of compliance with adjuvant therapy in modern series approximate 50% [133,139,173]. Furthermore, more recent data favor FLOT chemotherapy (docetaxel, oxaliplatin, leucovorin, fluorouracil), with demonstrated improved survival over the ECF/ECX regimen [133]. These findings, coupled with the results of the landmark MAGIC trial favoring delivery of neoadjuvant therapy [139], suggest that postoperative radiotherapy only has benefit in highly selected patients with significant adverse pathological features.

#### 4.8.2. Neoadjuvant Chemoradiation

A few series have investigated the benefit of neoadjuvant therapy in gastric cancer. Early investigations included patients with gastric cardia tumors comparing surgery alone to multimodal therapy consisting of fluorouracil and cisplatin with concurrent 40 Gy in 15 fractions [177]. With a limited median follow-up of 10 months, they reported an improved median survival with multimodality therapy to 16 months compared to 11 months; however, these overall outcomes poorly compared to contemporary results implicating inferior surgical technique. However, of note, the pCR rate in resected patients treated with multimodality therapy was 25% and lymph node positivity was reduced by nearly 50%. The phase II RTOG 9904 further investigated this strategy in 49 patients with gastric cancer, treating patients with induction fluorouracil/cisplatin followed by chemoradiation to 45 Gy with weekly fluorouracil and paclitaxel and by surgical resection (50% D2) [178]. Despite major protocol deviations in 56% of radiotherapy treatment and 37% of surgery, a pCR rate of 26% was achieved that was associated with per protocol treatment (*p* = 0.02). More modern data are extrapolated from the POET trial where, despite elevated post-op mortality with combined modality treatment to 30 Gy, there was a trend for improved survival at 5 years [98].

Overall, these findings have driven interest in larger randomized trials investigating this strategy specifically for gastric cancer. Of particular interest is CRITICS II comparing three arms of neoadjuvant therapy including: standard chemotherapy with DOC × 4 cycles (docetaxel, oxaliplatin, capecitabine), neoadjuvant DOC × 2 cycles followed by chemoradiation to 45 Gy with concurrent weekly carboplatin and paclitaxel, and chemoradiation alone with the winner to be further assessed in a phase III format [179]. Finally, early safety outcomes of the TOPGEAR trial have been reported comparing perioperative ECF chemotherapy for six cycles to preoperative ECF for two cycles followed by neoadjuvant 45 Gy with concurrent fluorouracil or capecitabine and adjuvant ECF for three cycles [180]. Compliance with neoadjuvant therapy in both arms has been >90% with preoperative treatment and >85% of patients are able to proceed to surgery. However, 65% of the chemotherapy alone arm and 53% in the chemoradiation group have completed postoperative chemotherapy. Importantly, there have been no significant differences in toxicity or surgical morbidity between the two treatment arms. In contrast to esophageal cancer, a lack of high-level evidence exists for neoadjuvant chemoradiation in gastric cancer while the final results of these studies are awaited.

#### 4.8.3. Unresectable Gastric Cancer

Inoperable patients or unresectable gastric cancer, including linitis plastica, are classically thought to be incurable; however, durable local control and an associated improvement in quality of life or palliation may be achieved with chemoradiation; see Figure 14 and Figure 15. Historic radiotherapy techniques had significant early mortality associated with chemoradiation to 50 Gy but improved 4-year survival of 17% vs. 6% with chemotherapy alone [181]. Similar outcomes have been noted in a retrospective analysis of the MD Anderson experience, demonstrating 5-year survival rates of 18% in 71 unresected patients with improved outcomes associated with chemoradiation (HR 0.25, 95%CI 0.06–1.01) with a median dose of 45 Gy (range: 36–50.4 Gy) [182]. At the population level, an analysis of the NCDB demonstrated a 7% improvement in 2-year survival with chemoradiation as opposed to chemotherapy alone [183]. Alternatively, there have been case reports of the use of proton therapy in Japan for inoperable patients treated to 61–83 Gy with at least one clinical complete response 2 years after treatment [184,185]. Undoubtedly, this is an area ripe for further inquiry as interest and utilization of particle therapy increases. The role of chemotherapy in the treatment of metastatic gastric cancer is very similar to esophageal cancer (see Section 3.3). Several of the metastatic esophageal adenocarcinoma trials included gastric cancer trials and vice versa.

#### 4.8.4. Gastric Neuroendocrine Tumors

The role of radiation therapy for neuroendocrine tumors is generally limited to palliation of symptomatic or oligometastatic disease. To this end, hypofractionated regimens offer patient convenience and the ability to achieve high biological equivalent doses to improve local control [186,187]. There is some limited evidence that SBRT may be an effective option for treatment of functional neuroendocrine tumors, with clinical and biochemical responses reported in four patients [188].

## 5. Future Directions in Diagnosis of Esophageal and Gastric Tumors

Novel methylated DNA markers assayed from plasma can detect esophageal cancer with moderate accuracy [189]. Serum markers to detect early esophageal and gastric cancer may be the first step to identifying patients at higher risk to undergo more invasive endoscopic screening. A non-endoscopic test, Cytosponge-trefoil factor 3 (TFF3), has demonstrated improved detection of Barrett esophagus compared with standard screening in a primary care setting [13]. Tethered capsule optical coherence tomography (OCT) is a non-endoscopic technique that has been developed and can volumetrically image esophageal mucosa and detect BE in unsedated patients in the outpatient setting [190].

In addition to advances in optical chromoendoscopy and endoscopic ultrasound, new modalities of image enhanced endoscopy are being studied. Techniques such as volumetric laser endomicroscopy (VLE) and confocal laser endomicroscopy have been shown to improve the detection of dysplasia associated with Barrett’s esophagus [191]. The addition of wide-area transepithelial sampling to an already thorough examination with HDWLE-NBI-VLE-SP may increase the yield of dysplasia detection. The future will likely involve using a combination of diagnostic methodologies to improve the accuracy of detection of preneoplastic lesions.

Endoscopic resection techniques such as ESD, submucosal tunnel endoscopic resection (STER), and full thickness resection devices (FTRD) will become more refined and the design of devices will likely improve as well, making this the primary treatment of choice for early-stage/mucosal cancer of the esophagus and the stomach.

There has been interest in developing and studying drug-eluting stents and stents combined with brachytherapy delivery for treatment and palliation of unresectable, obstructing gastrointestinal cancers. It is likely that we will see the development of such stents that facilitate delivery of brachytherapy or local chemotherapy for esophageal cancer in the future.

## 6. Chemotherapy and Radiation

In the last decade, a major breakthrough in cancer treatment was the discovery of immune checkpoint proteins, which effectively inhibit the immune system through various mechanisms. Immune checkpoint inhibitors are currently approved in several tumors in the metastatic and adjuvant setting. In the upper GI malignancies, approval is currently restricted to the metastatic setting and beyond the first line. Recent studies have shown promising activities in the front-line setting and in the adjuvant setting at least in a subset of patients and approval is expected in the upcoming months. Currently, several ongoing trials are evaluating the integration of immune checkpoint inhibitors in the neoadjuvant setting in combination with chemotherapy and radiation as well as following surgical resection. Combining immune checkpoint inhibitors with other targeted therapies (Her-2 and VEGF) has shown some promising activity as well and is currently being tested. Advances in image-guided radiation and proton beam therapy to improve precision are promising as well to decrease toxicity and improve patient tolerance.

Several new tyrosine kinase inhibitors have shown promising activity in GIST and they will be tested in the adjuvant and the neoadjuvant setting as well. Several novel agents have been approved for neuroendocrine tumors in the last 5 years and several treatment combination trials are ongoing.

## 7. Conclusions

Upper GI neoplasia is a complex process involving the organs of the upper digestive tract with a multifactorial etiology and represents a significant burden of disease worldwide. Diagnosis and staging require a multimodal approach, involving endoscopy, endoscopic ultrasonography, endoscopic mucosal resection and submucosal dissection techniques, and imaging CT/PET imaging modalities. Once diagnosis and staging are achieved, treatment involves careful multidisciplinary planning involving gastrointestinal endoscopists, medical oncologists, surgical oncologists, and radiation oncologists to determine the most appropriate therapeutic approach. For early lesions, therapeutic endoscopic approaches include endoscopic mucosal resection and endoscopic submucosal dissection. For locally advanced lesions, adjuvant or neoadjuvant chemotherapy therapy may be required based on staging, prior to surgical resection. Image-guided radiotherapy represents an important adjuvant or neoadjuvant therapy, again depending on stage and anatomy.

Future directions in screening may involve office-based tissue sampling strategies along with serological “liquid biopsy” assays. Advances in the diagnosis and staging of gastrointestinal malignancy will rely upon advances in endoscopic imaging, including optical coherence tomography and endoscopic ultrasound. Improvements in endosurgical techniques represent an area of promise for the therapy of early lesions, while advances in surgical oncology, novel chemotherapeutics, and tailored radiotherapy will hopefully allow for decreased disease morbidity and mortality over time.

## Figures and Tables

**Figure 1 cancers-13-00582-f001:**
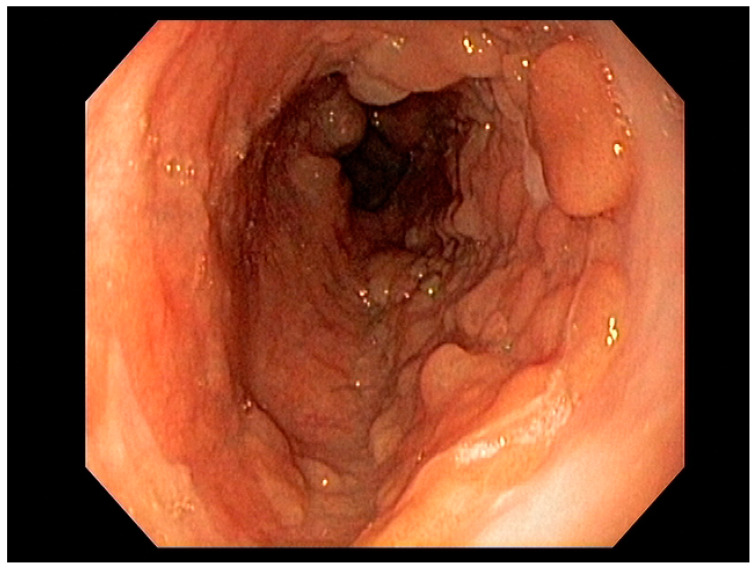
Barrett’s esophagus with Nodules White Light Endoscopy.

**Figure 2 cancers-13-00582-f002:**
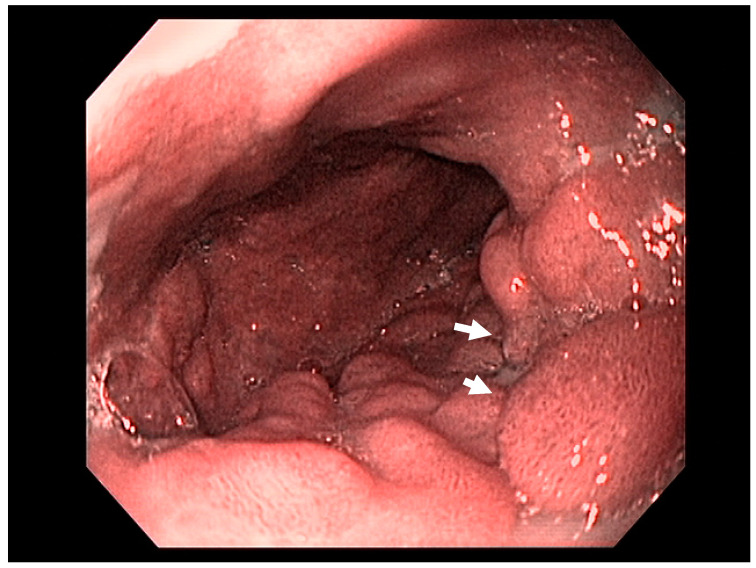
Barrett’s esophagus with nodules under narrow band imaging (NBI) (arrows).

**Figure 3 cancers-13-00582-f003:**
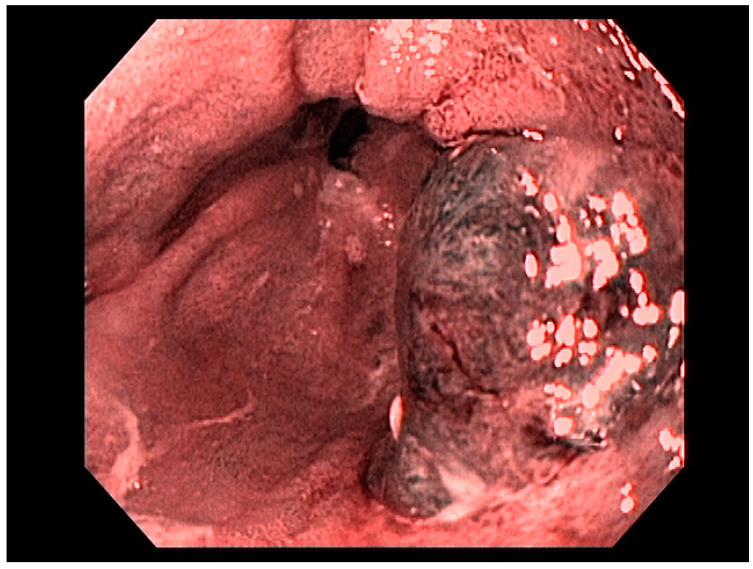
T1a esophageal cancer in Barrett’s Esophagus.

**Figure 4 cancers-13-00582-f004:**
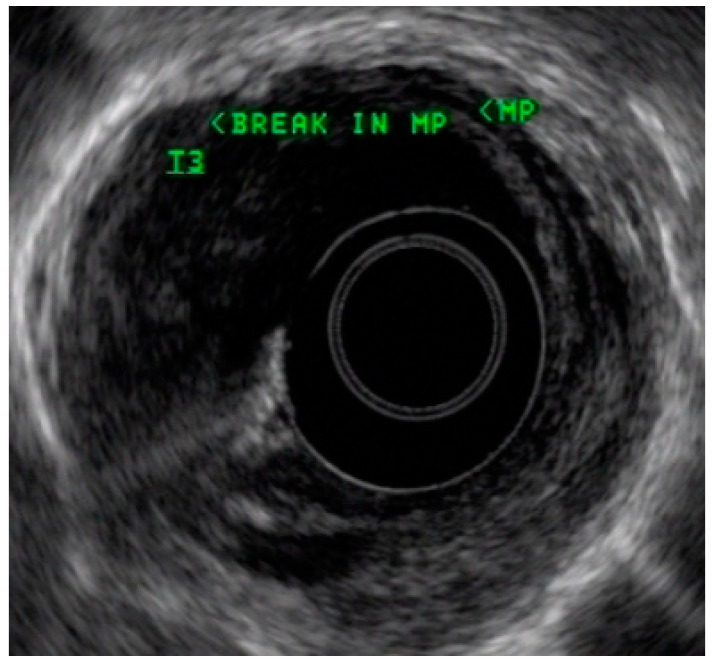
Esophageal cancer endoscopic ultrasound (EUS) image T3.

**Figure 5 cancers-13-00582-f005:**
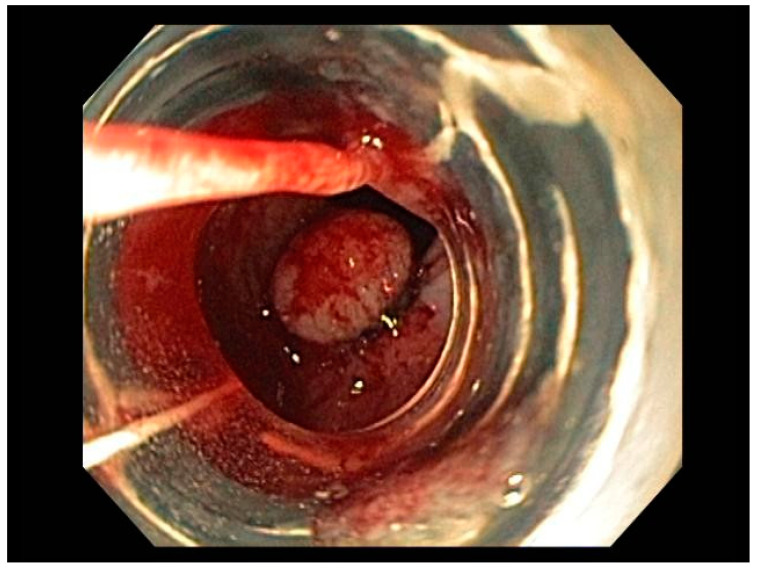
Band-assisted mucosectomy.

**Figure 6 cancers-13-00582-f006:**
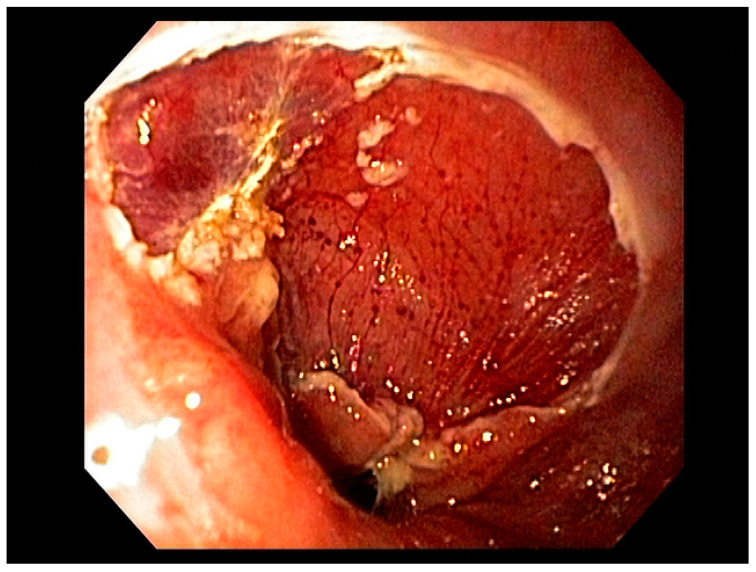
Endoscopic Mucosal Rejection (EMR) T1 esophageal cancer; resection site.

**Figure 7 cancers-13-00582-f007:**
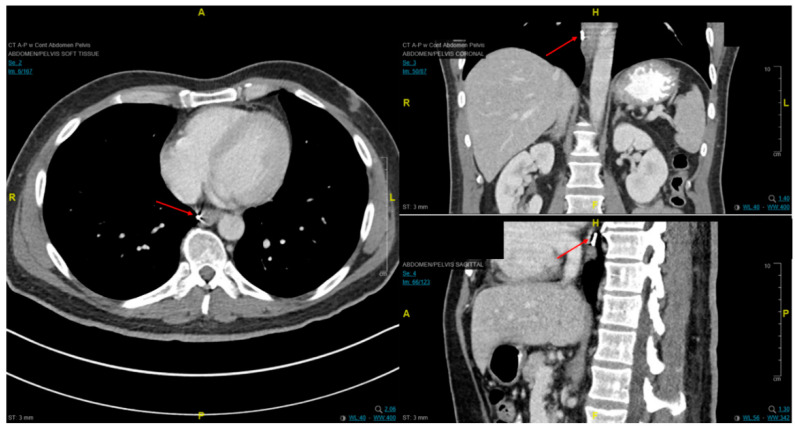
Axial, coronal, and sagittal CT images demonstrating EUS-placed fiducial (red arrow) proximal to esophageal tumor.

**Figure 8 cancers-13-00582-f008:**
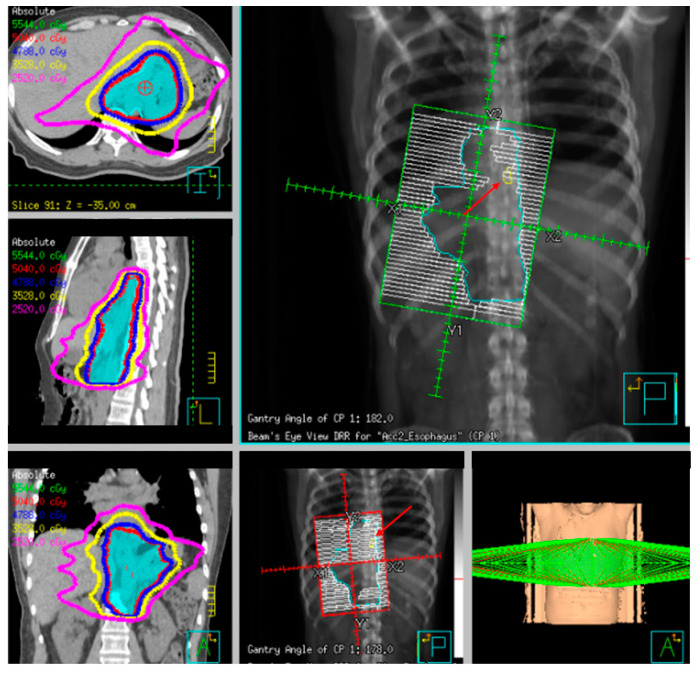
Example radiotherapy plan for image-guided treatment with fiducial contoured (red arrow) and within the planning target volume (PTV).

**Figure 9 cancers-13-00582-f009:**
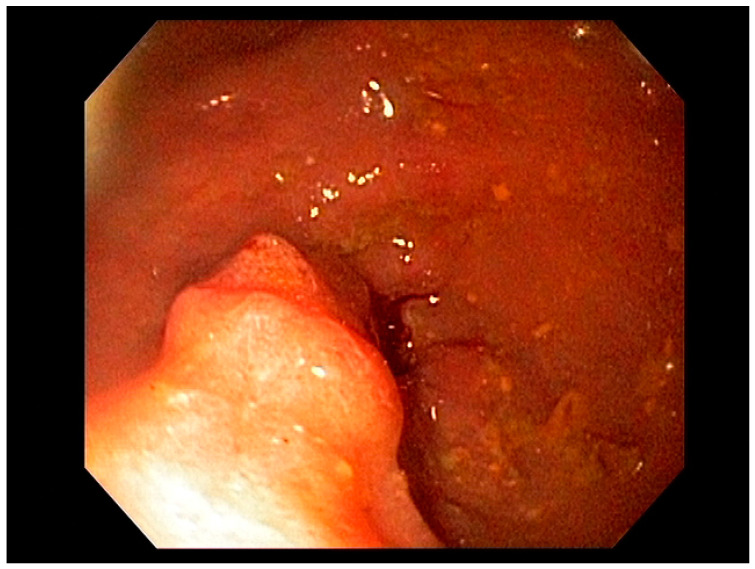
High-definition white light endoscopy (HD-WLE) combined with optical chromoendoscopy techniques such as narrow-band imaging (NBI) and targeted biopsies.

**Figure 10 cancers-13-00582-f010:**
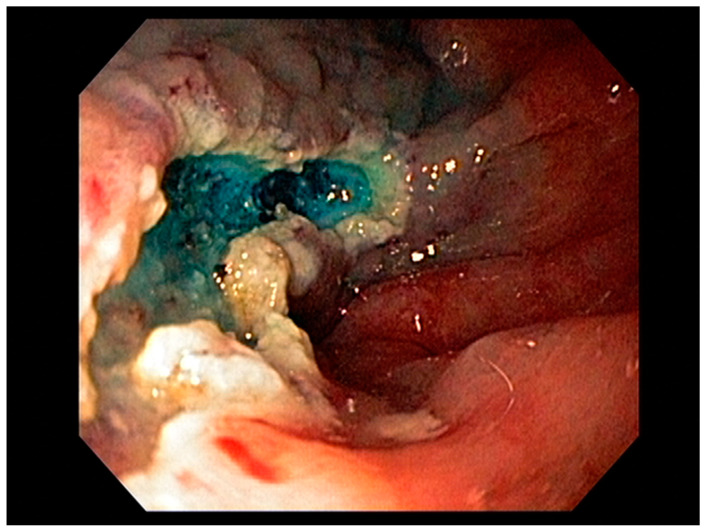
EMR of Early Gastric Cancer (EGC); submucosa stained with methylene blue.

**Figure 11 cancers-13-00582-f011:**
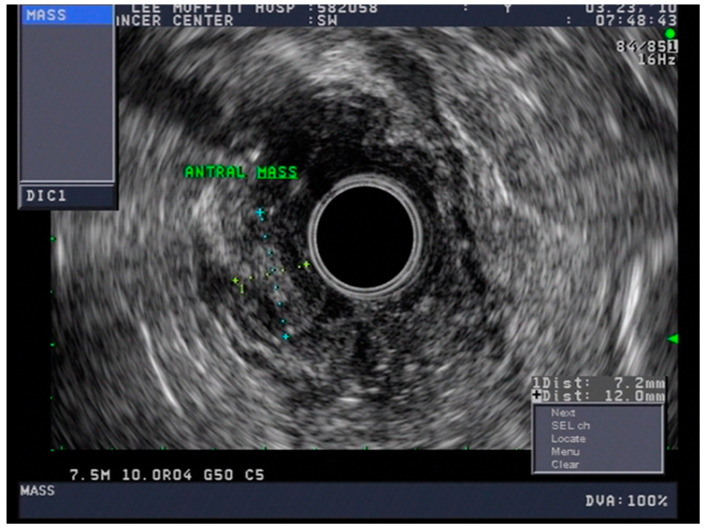
Early gastric cancer; EUS image polypoid lesion T1N0.

**Figure 12 cancers-13-00582-f012:**
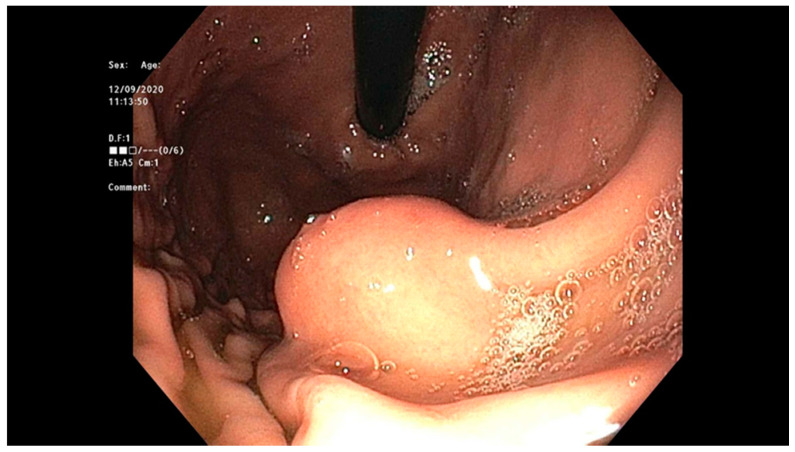
Gastrointestinal stromal tumor, as seen with high-definition white light endoscopy.

**Figure 13 cancers-13-00582-f013:**
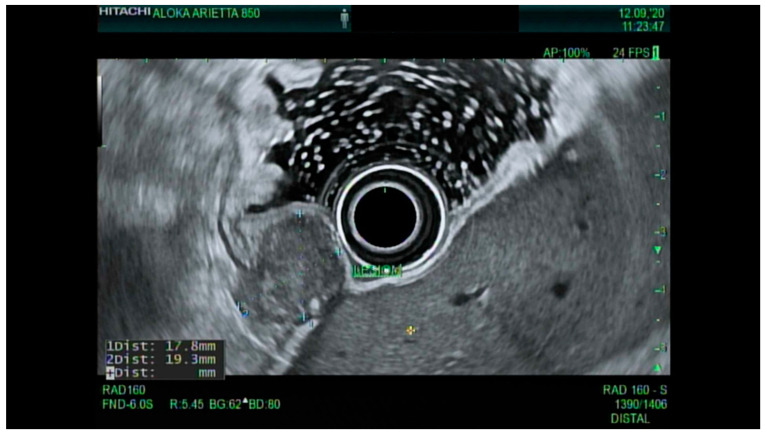
EUS images of a gastrointestinal stromal tumor originating from muscularis propria.

**Figure 14 cancers-13-00582-f014:**
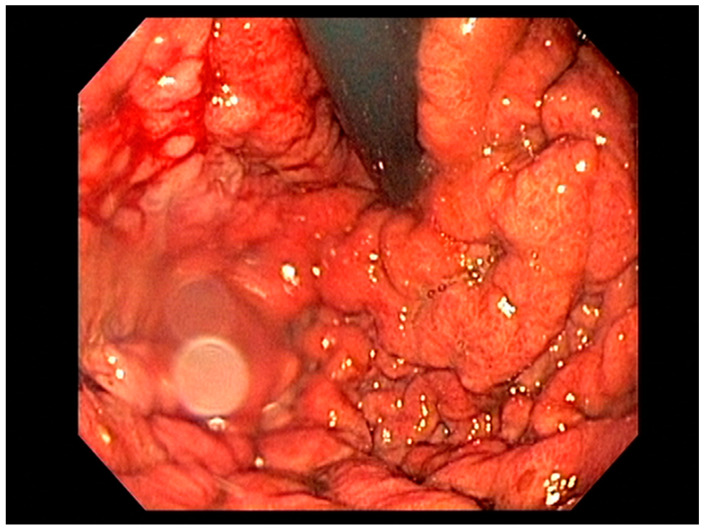
Gastric linitis plastica, submucosal infiltrating cancer; white light endoscopy.

**Figure 15 cancers-13-00582-f015:**
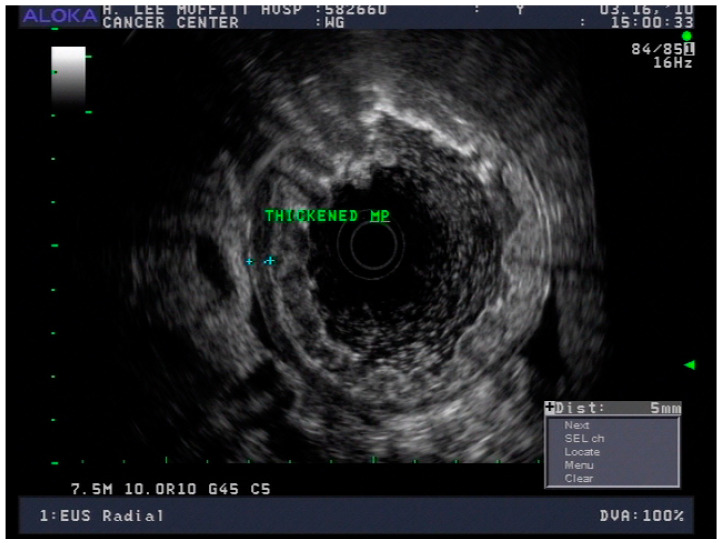
EUS Linitis plastica with significant wall thickening.

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
