# Peer review of "Epidemiology, Diagnosis, Staging and Multimodal Therapy of Esophageal and Gastric Tumors"

_cancers, 2021, doi:10.3390/cancers13030582_

Round 1

Reviewer 1 Report

Summary

In this article, authors summarized the epidemiology, diagnosis, staging and multimodal therapy of Esophageal and Gastric Tumors. This review article is very well organized and will be very helpful for the readers.

These findings are very intriguing and this study will meet the criteria of this journal. However, there were several critical points to be corrected to meet the criteria of this journal.

Query

  1. In this review, chemoradiotherapy was mentioned, but only a little about chemotherapy. The drug development for GI cancers is very active recently, authors should mention the situation. Recently, there are some reports of combination drug therapy especially molecular target agents or immune check point agents.
  2. Although minimal invasive surgery prevails, there is no discussion about minimal invasive surgery for esophageal and gastric cancer such as robotic assisted surgery and thoracoscopic surgery.
  3. Recently, Laparoscopy and Endoscopy Cooperative Surgery (LECS) was performed for GIST, authors should mention about it.
  4. Figure2 was unclear. The authors should carry the more clearly.
  5. The Japanese gastric cancer guideline 3rd edition was outdated. The latest is Japanese gastric cancer treatment guidelines 2018 (5th edition). Gastric cancer
  6. Page14-15 “Early Gastric Cancer: Endoscopic diagnosis and therapy” section, the authors should mention for undifferentiated EGC <20mm
  7. Page 9, Line 291, Ref60 is not superscripted and should be corrected.

Author Response

Thank you for considering our article for your journal and your feedback. Below, please find our responses to the specific concerns raised by reviewer # 1.

Query

  1. In this review, chemoradiotherapy was mentioned, but only a little about chemotherapy. The drug development for GI cancers is very active recently, authors should mention the situation. Recently, there are some reports of combination drug therapy especially molecular target agents or immune check point agents.      

  • Thank you for this valuable input. Additional information has been added about chemotherapy for gastric malignancies, particularly with regard to novel neoadjuvant and adjuvant chemotherapeutic regimens are discussed, including details on molecular and immunotherapies.

2.  Although minimal invasive surgery prevails, there is no discussion about minimal invasive surgery for esophageal and gastric cancer such as robotic assisted surgery and thoracoscopic surgery. Recently, Laparoscopy and Endoscopy Cooperative Surgery (LECS) was performed for GIST, authors should mention about it.

  • Thank you for this valuable input. Additional information has been included to discuss minimally invasive surgical approaches including laparoscopy, robotic surgery particularly with therapy for esophageal malignancies.

3. Figure2 was unclear. The authors should carry the more clearly.

  • Thank you for this comment, however the feedback was unclear regarding issues with figure 2. All figures have been uploaded with the highest graphic resolution available. The caption for figure 2 has been modified for clarity. The latest gastric cancer guidelines have now been referenced.

4. The Japanese gastric cancer guideline 3rd edition was outdated. The latest is Japanese gastric cancer treatment guidelines 2018 (5th edition). Gastric cancer.

  • Thank you for this comment, the latest gastric cancer guidelines have now been referenced.

5. Page14-15 “Early Gastric Cancer: Endoscopic diagnosis and therapy” section, the authors should mention for undifferentiated EGC <20mm.

  • Thank you for this valuable input. We have included additional information about diagnosis, staging, and therapy of undifferentiated gastric cancers less than 20mm, including role and appropriateness of endoscopic resection techniques.

6. Page 9, Line 291, Ref60 is not superscripted and should be corrected.

Thank you for your input. We have corrected this formatting error.

Reviewer 2 Report

The title of this review article is very ambitious. Unfortunately, the manuscript does not satisfy the aim of this title.

The work is too superficial and too fragmented to provide a careful and exhaustive lecture on tumors from esophagus to ligament of Treitz.     

Author Response

We appreciate the reviewers feedback. While we recognize this is an ambitious work, our goal is more to highlight the role of multidisciplinary care for treatment of these lesions. We have focused on common tumors of the esophagus and stomach. We anticipate with our additional edits per the other reviewers comments we have provided a solid review on this topic.  

Reviewer 3 Report

First of all I would like to congratulate the authors for their well-designed work. This work offers a comprehensive information for esophageal and gastric tumours. I strongly believe that this work represents a valid review for the treatment of these malignancies. The need of multimodal treatment is sufficiently highlighted. 

Author Response

We thank the reviewer for his kind feedback. We are glad to note that this manuscript is comprehensive and a good review of esophageal and gastric tumors based on his review. We have reviewed the manuscript again for spelling and grammar. 

Round 2

Reviewer 2 Report

No further comment